# Effective communication and collaboration with health professionals: A qualitative study of primary care pharmacists in Western Australia

Tin Fei Sim[1]*, H Laetitia Hattingh[1,2,3], Bruce Sunderland[1], Petra Czarniak[1]

1 School of Pharmacy and Biomedical Sciences, Faculty of Health Sciences, Curtin University, Bentley, Western Australia, Australia, 2 Gold Coast Health, Gold Coast, Queensland, Australia, 3 School of Pharmacy and Pharmacology, Griffith University, Gold Coast, Queensland, Australia

* T.Sim@curtin.edu.au

## Abstract

### Background

The expanded provision of professional services by community pharmacists in the primary care setting encompasses the necessity to communicate and collaborate with other health professionals. Little is currently known about contemporary processes employed for their achievement.

### Objective

To explore contemporary processes employed for effective communication and collaboration between primary care pharmacists and health professionals.

### Methods

In-depth semi-structured interviews were conducted with pharmacists practising in primary care settings requiring varying expertise, practice experience and speciality backgrounds. Interviews were audio-recorded, transcribed verbatim and coded using NVivo version 11. Data were analysed following an inductive approach to facilitate thematic analysis.

### Results

Twenty-six pharmacists were interviewed, which achieved data saturation. Five overarching themes emerged as participants described their experiences and perspectives regarding processes employed for communication and collaboration: i) tailored means of communication, ii) referral processes, iii) facilitators for effective interactions, iv) barriers to effective interactions, and v) implementation of a national digital health record. Participants acknowledged that the changing landscape of the Australian health system affected communication and collaboration with other health professionals. The changes resulted in participants' acceptance of a multidisciplinary approach to healthcare, which was contingent upon effective communication, interactions and relationships with other health professionals. Varying

**Funding:** This project was funded by a John M O'Hara Research Grant, Pharmaceutical Society of Western Australia, Australia. The funder had no role in study design, data collection and analysis, decision to publish, or preparation of the manuscript.

**Competing interests:** The authors have declared that no competing interests exist.

levels of formality and characteristics of referrals were identified, however the nature of the communication was tailored to the individual scenario or circumstance that was considered appropriate.

## Conclusions

Pharmacists exercised judgment on a case-by-case basis when tailoring the means of communication. The establishment of a consistent and structured two-way referral process between health professionals within the primary care setting is important, which includes use of the national digital health record. Increased awareness and appreciation of each health professional's roles and expertise would further enhance inter-professional collaboration.

## Introduction

Pharmacy practice in Australia, in line with international trends, has evolved from a focus on dispensing and supply towards a growing emphasis on the provision of professional and cognitive services [1–6]. There appears to be no consistent definition which encompasses the range of services offered by pharmacists practising in primary care settings [7]. Professional pharmacy services are considered as those "contributing to patient health, through effective interaction with both patients and other health professionals" [2].

The Australian primary health system has undergone significant changes over the last decade, many of which have had a direct influence on community pharmacy practice. The Pharmaceutical Benefits Scheme (PBS) has been a long-standing platform for the provision of a wide range of medicines readily accessible to the public from community pharmacies, subsidised by the Australian Government [8, 9]. Price disclosure refers to a program within the PBS which "progressively reduces the price of some PBS medicines which are subject to competition, ensuring better value for money from these medicines" [8]. This change has reduced pharmacist remuneration from the PBS, but has been coupled with a positive trend in government funding for professional services through several Community Pharmacy Agreements [10]. The result has contributed to a paradigm shift in community pharmacy practice. The increased provision of professional services by pharmacists augments the requirements to interact with and involve patients through shared decision-making. It also increases the obligation for communication and collaboration with other healthcare professionals who are involved in the care of the patients. A perceived increase in interactions between pharmacists and general practitioners (GPs) through the provision of professional pharmacy services has also been identified in previous Australian studies [11, 12].

As part of the paradigm shift, defined career pathways where pharmacists are increasingly embedded within other primary care practice settings, have been established. These include working alongside other health professionals, including GPs, in residential aged care facilities (RACFs) and Aboriginal Health Services (AHSs) [13, 14]. In addition to GPs, close-working relationships between community pharmacists and other health professionals, either co-located within, or independent of a community pharmacy, have become a more common phenomenon in primary care. This includes interactions with nurse practitioners [15–17] and psychologists [18, 19]. Accredited pharmacists similarly have regular interactions with other health professionals. In Australia, these pharmacists are accredited with either the Australian Association of Consultant Pharmacy (AACP) [20] or the Society of Hospital Pharmacists

Australia (SHPA) [21]. They conduct Australian government-funded medication reviews for patients in their home or for residents of RACFs. Although the Pharmacy Board of Australia supports and has declared no regulatory barriers for pharmacists to prescribe under supervision or a structured collaborative arrangement [22], pharmacists in Australia do not currently have prescribing rights resembling the models in the United Kingdom (UK) [23] and Canada [24]. There is also a lack of a formal and remunerated referral process in place for Australian community pharmacists in relation to the management of minor ailments similar to those associated with the Minor Ailments Scheme in the UK [25].

A multidisciplinary approach to patient care involves effective interactions, communication and collaboration, including referrals of patients, between health professionals throughout the continuum of care. This involves sharing patients' health information in a secure, reliable and timely manner [26, 27]. The evolving health system and momentum for inter-professional collaboration in Australia has led to the establishment of a national digital health record system, now known as the "My Health Record", an initiative of the Australian Government's Australian Digital Health Agency [26]. This system provides an online summary of key health information of people who live in Australia, unless they choose not to have a record [28]. With patient consent, health professionals including pharmacists, are able to access, view and add information to a patient's record. The implementation of this system in 2019 represented an additional means of 'communication' between health professionals. The effectiveness and risk management of the implementation of this system under the opt-out model was evaluated by the Australian National Audit Office (ANAO) [29]. It reported the implementation was considered "effective" and "appropriate". The report further highlighted the robust controls which were put in place to enable cyber security and privacy, although on-going regular monitoring for compliance with access and security legislation was recommended. The number of active records has increased by 190,000 since July 2019 to a total of 22.71 million. Following implementation campaigns, over 90% of community pharmacies and general practices are registered with the system [30].

A systematic review conducted by Mahdizadeh et al. reviewed frameworks and models of clinical interdisciplinary collaboration published in the literature between 1990 and 2014 [31]. Although this review focused on collaboration between nurses and medical doctors, it highlighted the roles of organisational structure, social and cultural factors in facilitating clinical collaboration between these professionals [31]. A German study explored the perceptions of inter-professional communication in relation to methods and contents of communication [32]. The study found that although differences in opinions occurred between GPs and pharmacists, communication was perceived to be crucial and that any future recommendations or models should allow flexibility for these practitioners to make adjustments to suit their unique needs [32]. An Australian study conducted between 2013 and 2014 proposed a model to facilitate collaboration between pharmacists and GPs in the context of addressing medication adherence [33]. The model highlighted elements of "shared pragmatic perspectives and trust" and "regular, face-to-face interactions" [33]. Nevertheless, there has been significant progress over the last five years in relation to digital health and the provision of cognitive professional pharmacy services which may influence interactions and communication between health professionals. Currently available models also do not address pharmacists' interactions with a diverse group of health professionals, including GPs, nurses and other allied health professionals in a contemporary practice landscape.

Considering the expanding roles of Australia's pharmacists working in primary care in a rapidly changing practice landscape and the increased requirement for inter-professional interactions, little is known about current processes which lead to effective communication

and collaboration. An understanding of these processes should lead to developing systems that enhance interdisciplinary collaboration and patient care.

This study aimed to explore contemporary processes to achieve effective communication and collaboration between community pharmacists and other health professionals, through exploring pharmacists' views and experiences in the context of a changing practice landscape.

## Materials and methods

This study was reviewed and approved by the Curtin University Human Research Ethics Committee (HREC2017-0036-02). All participants provided written informed consent before participating in interviews. The research team consisted of four academic researchers, three females and one male, and were all pharmacists with academic positions at an Australian university. All four research team members have prior experience in qualitative and pharmacy practice research.

### Design and setting

The design of the study and the presentation of findings followed the Consolidated Criteria for Reporting Qualitative Research (COREQ) 32-item checklist [34]. This study involved semi-structured interviews with Western Australian pharmacists from diverse and varying practice backgrounds, to gain an in-depth understanding of the topic. This manuscript focuses on reporting detailed demographic information about the participants, their practice settings and professional roles, and processes used to achieve effective communication and collaboration with other health professionals in the context of a changing practice landscape. Pharmacists' perspectives on the implementation of services, including prior investigation and research undertaken, their considerations and perceptions of the impact of services, and factors influencing the success of service provision, are reported elsewhere [6].

### Participants

Considering the expansion of scope and role of pharmacists, this study purposively selected pharmacists practising in Western Australia (WA) primary care settings requiring varying expertise and practice backgrounds, including community and accredited pharmacists with additional qualifications. Purposive sampling allowed maximum variation, thus enabling in-depth inquiry into a range of professional services provided by pharmacists in the current practice landscape [35]. Through the researchers' network and consultation with pharmacy professional bodies, an initial list of prospective participants was developed. In addition to practice backgrounds and experience, participants' willingness and availability to participate as well as articulation and communication skills were taken into consideration. Subsequent participants were recruited via the snowballing method [36]. The prospective participants were contacted and invited to participate in the study via email or telephone between September 2017 and February 2018. This purposive recruitment method enabled pharmacists with additional qualifications to provide extended services such as pharmacist-administered immunisation services, diabetes education, wound management, integrated healthcare and specialised compounding to be included. The diverse backgrounds and characteristics of participants enabled an information-rich discussion.

### Interview tool and data collection

A semi-structured interview tool comprising of six parts: Part A–pharmacist demographic data, Part B–current professional roles, Part C–community pharmacists, Part D–accredited

pharmacists, Part E–diabetes educators and pharmacists providing other specialities, and Part F–general opinion, was developed to guide all interviews. It was validated by four academic pharmacists for face and content validation, and trialled under intended interview conditions with two practising pharmacists for the purposes of confirming validity and as practice for the interviewer. Data from these trial interviews did not contribute towards subsequent analysis. To avoid bias and assumptions, all interviews were conducted by an independent interviewer who was a female pharmacist with a diverse practice background and previous skill in undertaking interviews. The practice background and experience of the interviewer allowed digression of the interview conversation as appropriate to enable information-rich data to be collected. In addition, the interviewer did not take part in the design of the study, construction of the interview tool or data analysis. All interviews were conducted via telephone and audio-recorded, with participants' consent, then transcribed verbatim. Transcripts were de-identified by replacing participants' identities with specific codes which represented their practice settings and qualifications. For example, "P01 CA" denoted participant number 1 who was a community (C) and accredited (A) pharmacist; whereas "P23 CAIS" denoted participant number 23 who was a community (C) and accredited (A) pharmacist with additional credentialing to undertake immunisation (I) and other specialised (S) services. All participants were provided with a participant information statement with details about the study, including the background and study objectives and contact details of the research team.

## Data analysis

De-identified transcripts were imported into NVivo version 11 (QSR International Pty Ltd) for organisation of the data and subsequent data analysis. Two researchers (TFS and LH), both pharmacist academics with experience in qualitative research, analysed the data independently following an inductive approach to facilitate thematic analysis [36, 37]. Transcripts were read repeatedly by the researchers to gain a deep understanding of the topics discussed, before initial ideas were coded as 'nodes' under an initial framework for coding. Codes were then grouped to form categories, which subsequently formed sub-themes, addressing the processes for effective communication and collaboration between pharmacists and other health professionals (reported in this paper). Data saturation was considered when no new codes were identified during the coding process. During the data analysis phase, the two researchers had regular meetings to openly discuss any disagreements to reach consensus. When consensus was not reached, the other two researchers were involved to resolve any disagreements. Other data relevant to factors influencing the implementation and maintenance of professional pharmacy services were reported elsewhere [6].

## Results

A total of 32 pharmacists were approached and invited to participate, with 26 (16 males; 10 females) subsequently consenting to participate in the semi-structured interviews. The reason for declining to participate was difficulty in finding a mutually suitable time to undertake the interview. The researchers were confident that data saturation was reached after 26 interviews as no new themes emerged. At the time of interview, all participants were registered with the Australian Health Practitioner Regulation Agency (AHPRA) and practising in WA primary care settings as a pharmacist. Table 1 provides a detailed summary of the demographic characteristics of participants, their practice settings and professional roles.

Of the 26 participants, 20 were practising in a community pharmacy setting whilst 13 were pharmacists accredited with the Australian Association of Consultant Pharmacy (AACP). Seven participants were practising as both a community and accredited pharmacists. All

Table 1. Demographic characteristics of participants and their practice settings (n = 26).

| Code* | Gender | Practice setting (size# and location) | Pharmacy type (banner or independent) | Experience as a pharmacist (years) | Postgraduate qualification/ certification | Professional roles | Accreditation status€ |
|---|---|---|---|---|---|---|---|
| ¥ P01 CA | Female | Small, rural community pharmacy | Independent | 16 | AACP accreditation | Proprietor, manager and pharmacist of a community pharmacy, accredited to conduct medication reviews | Accredited pharmacist servicing rural WA regions |
| P02 CI | Male | Large, metropolitan community pharmacy | Banner | 6.5 | Immunisation | Manager and pharmacist of a community pharmacy, administers immunisation service | N/A |
| P03 C | Male | Small, rural community pharmacy | Independent | 7 | N/A | Manager and pharmacist of a community pharmacy | N/A |
| P04 C | Male | Large, metropolitan community pharmacy | Independent | 31 | N/A | Manager and pharmacist of a community pharmacy which provides mental health services | N/A |
| ¥ P05 CAIS | Female | Large, metropolitan community pharmacy, university | Independent | 28 | AACP accreditation, immunisation, lactation consultancy, asthma education, Diploma in Advanced Dementia Care, integrative health | Proprietor, manager and pharmacist of a community pharmacy, sessional academic/ teaching at local university, administers immunisation services, practices integrative medicines, serves on pharmacy-related committees | Accredited pharmacist servicing metropolitan region |
| P06 C | Male | Large, metropolitan community pharmacy | Independent | 32 | N/A | Proprietor and pharmacist in a community pharmacy, non-executive director roles on private, listed, and not-for-profit organisations, serves on industry and government-related advisory committees | N/A |
| P07 C | Male | Large, metropolitan community pharmacy | Banner | 24 | N/A | Management role and pharmacist within a banner group with a discount model | N/A |
| P08 C | Male | Large, metropolitan community pharmacy | Banner | 23 | N/A | Proprietor and manager of a community pharmacy | N/A |
| P09 A | Female | Home visits, metropolitan hospital, university | N/A | 8 | AACP accreditation | Accredited to conduct medication reviews, hospital pharmacist and sessional academic/teaching | Accredited pharmacist servicing metropolitan region |
| P10 A | Male | Home visits, residential aged care facilities and university | N/A | 15 | AACP accreditation, PhD in Pharmacy | Accredited to conduct medication reviews, management role in an organisation funded by the Federal Government | Accredited pharmacist servicing metropolitan region |
| ¥ P11 CAI | Male | Small, rural community pharmacy | Banner | 7 | AACP accreditation, immunisation | Accredited to conduct medication reviews, pharmacist in a community pharmacy, administers immunisation services | Accredited pharmacist servicing rural WA regions |
| P12 CI | Male | Medium, metropolitan community pharmacy | Independent | 7 | Immunisation | Manager and pharmacist of a community pharmacy, administers immunisation services | N/A |
| ¥ P13 CAI | Male | Medium, metropolitan community pharmacy | Banner | 18 | AACP accreditation, Diploma in Clinical Pharmacy, immunisation | Proprietor, manager and pharmacist of a community pharmacy, accredited to conduct medication reviews, practice manager of a medical centre, administers immunisation services | Accredited pharmacist servicing metropolitan region |

(Continued)

**Table 1.** (Continued)

| Code* | Gender | Practice setting (size# and location) | Pharmacy type (banner or independent) | Experience as a pharmacist (years) | Postgraduate qualification/ certification | Professional roles | Accreditation status€ |
|---|---|---|---|---|---|---|---|
| ¥ P14 CAS | Female | Small, metropolitan community pharmacy, and general practice | Banner | 8 | AACP accreditation, Diploma of Management, Graduate Certificate in Wound Care | Pharmacist specialising in wound care services in a community pharmacy, accredited to conduct medication reviews, practises as a practice pharmacist in a general practice | Accredited pharmacist servicing metropolitan region |
| P15 CI | Male | Large, metropolitan community pharmacy | Independent | 18 | Immunisation | Proprietor and pharmacist of a community pharmacy, administers immunisation and sleep apnoea services | N/A |
| P16 CI | Male | Medium, metropolitan community pharmacy | Independent | 22 | Immunisation | Proprietor, manager and pharmacist of a community pharmacy, administers immunisation services | N/A |
| P17 AIS | Female | Home visits, residential aged care facilities, an independent private health clinic and general practice | N/A | 18 | AACP accreditation, Graduate Certificate in Diabetes Education (credentialed), Asthma education, immunisation | Accredited to conduct medication reviews, diabetes and asthma educator practising in own private clinic and a general practice | Accredited pharmacist servicing metropolitan region |
| P18 A | Male | Home visits and residential aged care facilities | N/A | 37 | AACP accreditation, Master of Clinical Pharmacy, Graduate Certificate in Diabetes Education | Accredited to conduct medication reviews, mainly in residential aged care facilities, provides clinical education to nurses and doctors | Accredited pharmacist servicing metropolitan region |
| P19 A | Female | Rural Aboriginal Health Service | N/A | 15.5 | AACP accreditation | Accredited to conduct medication reviews, mainly for the ATSI population in the rural communities of WA, provides education and training to staff and carers of patients | Accredited pharmacist servicing rural WA regions |
| P20 C | Female | Small, metropolitan community pharmacy | Independent | 9.5 | N/A | Proprietor, manager and pharmacist of a community pharmacy | N/A |
| P21 A | Female | Home visits, residential aged care facilities, general practice | N/A | 15 | AACP accreditation | Accredited to conduct medication reviews, practises in a general practice | Accredited pharmacist servicing metropolitan region |
| P22 CIS | Male | Large, metropolitan community pharmacy | Independent | 12 | Immunisation, specialised compounding, integration (with ACNEM), anti-ageing medicine (with A5M) | Pharmacist in a community pharmacy, provides specialised compounding, integrative medicine and immunisation services | N/A |
| ¥ P23 CAIS | Female | Medium, metropolitan community pharmacy | Independent | 17 | AACP accreditation, immunisation, specialised compounding | Proprietor, manager and pharmacist of a community pharmacy, provides specialised compounding and immunisation services, accredited to conduct medication reviews | Accredited pharmacist servicing metropolitan region |
| ¥ P24 CAS | Female | Home visits and general practice, rural community pharmacy (occasionally as a locum) | Various (as a locum) | 25 | AACP accreditation, Graduate Certificate in Diabetes Education (credentialed) | Accredited to conduct medication reviews, provides diabetes education, locum pharmacist in rural community pharmacies, practises in a general practice | Accredited pharmacist servicing rural WA regions |

(*Continued*)

**Table 1.** (Continued)

| Code* | Gender | Practice setting (size# and location) | Pharmacy type (banner or independent) | Experience as a pharmacist (years) | Postgraduate qualification/ certification | Professional roles | Accreditation status€ |
|---|---|---|---|---|---|---|---|
| P25 CI | Male | Medium, rural community pharmacy, Aboriginal Health Service | Independent | 6 | Immunisation | Manager and pharmacist of a community pharmacy, provides immunisation services and services to the Aboriginal health services | N/A |
| P26 CS | Male | Large, rural community pharmacy | Banner | 17 | Specialised compounding | Manager and pharmacist of a community pharmacy, provides specialised compounding services | N/A |

* C: Community pharmacist; A: Accredited pharmacist; I: Immuniser pharmacist; S: pharmacist providing Specialist services which require credentialing/certification

# Size of community pharmacy based on approximate gross turnover: small (< AUD 1.5 mil pa); medium (AUD 1.5–3.5 mil pa); large (> AUD 3.5 mil pa)

¥ Note: some accredited pharmacists were also practising in a community pharmacy at the time of the interview.

€ Pharmacists accredited with either the Australian Association of Consultant Pharmacy (AACP) or the Society of Hospital Pharmacists Australia (SHPA) to conduct federal government-funded medication reviews.

ACNEM: Australian College of Nutritional and Environmental Medicine; A5M: The Australasian Academy of Anti-Ageing Medicine; ATSI: Aboriginal and Torres Straits Islander; N/A: Not applicable; PhD: Doctor of Philosophy; WA: Western Australia.

participants were engaged in mixed roles, ranging from a practitioner (i.e. pharmacist on duty in a community pharmacy or performing medication reviews as an accredited pharmacist), to an administrative/leadership role (i.e. proprietor, manager or a director of a company). Participants also had varied qualifications and practice experience in specialist services, including immunisation, diabetes education, lactation consultancy, asthma education, specialised compounding, integrative health, wound care, dementia care and business management. All participants worked on average at least full time hours (i.e. more than or equivalent to 37.5 hours per week), with some working up to 60 hours per week.

Five overarching themes emerged when participants described their experiences and perspectives about processes to achieve effective communication and collaboration with other health professionals: i) tailored means of communication, ii) referral processes, iii) facilitators for effective interactions, iv) barriers to effective interactions, and v) implementation of a national digital health record.

## i) Tailored means of communication

It was noteworthy that participants recognised and acknowledged the changing landscape in pharmacy practice, driven by consumer needs and demands, and the evolving health system in Australia. These changes were also reported to be a major contributor to the increased need to work collaboratively with other health professionals. Throughout the interviews, participants identified and discussed various scenarios that involved interactions and communication with other health professionals. Other health professionals who were identified as having close working relationships with the participants included AHPRA-registered practitioners such as GPs, podiatrists, dieticians, physiotherapists, nurse practitioners, dentists, podiatrists, clinical psychologists, exercise physiologists and nurses in RACFs, and some practitioners who were diabetes educators. Of all the aforementioned health professionals, GPs were the most commonly mentioned.

Having an established diverse communication means or system, which enabled operational efficiency, was identified to have a major impact on the success of developing and maintaining

an on-going positive collaborative working relationship with other health professionals. During the interviews, participants described a range of distinctive methods of communicating with various health professionals, and the importance of developing and tailoring a method of communication that worked for both parties. It was important to note that the communication method may have differed dependent on the practitioner with which the pharmacist communicated. Having an established and consistent communication system with an individual practitioner was deemed to greatly enhance collaboration as practitioners were 'used to' being contacted in that manner.

Throughout the interviews, participants commented on various and diverse methods of communication, which appeared to be effective based on their experiences: *"Case conferences. . . phone calls, emails, face-to-face meetings, reports. . . don't use social media, but aside from that, most other forms of modern communication. . ."* (P10 A).

The nature and intended complexity of the interaction was also raised as a point for consideration. For example, if the communication was intended to be brief and straight-forward, a phone call was considered the most effective way: *". . .sometimes I might talk to a podiatrist or a dietician or someone like that if it's relevant. I would just ring them up."* (P05 CAIS). Some participants further explained that they would contact a prescriber directly by phoning the prescriber's mobile phone instead of phoning the practice, demonstrating a direct and efficient link to the prescriber. For other interactions especially in relation to medication reviews where more complex clinical recommendations were involved, written communication with clear documentation that enabled follow up were the preferred communication means. Specifically, one participant who provided Home Medicines Reviews articulated the process of two-way written communication with the prescriber:

*". . .the reports that we send. They usually comment on the report. So I usually leave a space for them to provide their medication management plan, and comment on their recommendations. And once they've done that they fax it back to me so then there is this communication going on about what has been done and what has, yes, been implemented."*

(P14 CAS).

Where suitable, regularly scheduled face-to-face meetings were also raised by some participants as an effective way to facilitate interactions especially if the interaction was likely to involve lengthy discussion and deliberation:

*"We basically have monthly clinical meetings at the medical centre, and we also have informal meetings that happen on a regular basis, at least once or twice a week. . . With the dental [practice], again, once in six months but again, at the same time, informal meetings can happen anytime. With the podiatrist, once in 12 months. And with the clinical psychologist, it would probably be once a month."*

(P20 C).

In addition to direct interactions, indirect communication with other clinics and practices that aimed to raise other health professionals' awareness of the services provided by the pharmacists was described as another means of communication, as described by one:

*"We basically sat in on their [general practices] clinical meeting for the week, and we sort of were able to present to them. We've also started sending monthly newsletters. . . monthly or*

*bi-monthly newsletters to our GP practices as well, just explaining any sort of new services that we're doing."*

(P26 CS)

## ii) Referral processes

Participants had varying interpretations of what constituted a referral. Participants' perceptions about referrals can be described as being on a spectrum of formality, varying from the least formal (e.g. verbally telling the patient to see a health professional), to increasing formality (e.g. contacting and making an appointment on behalf of the patient), to most formal involving documentation (e.g. writing a referral letter or note). This is illustrated in Fig 1.

Specifically, participant P06 C clearly narrated an experience in effective two-way referral between pharmacists and prescribers involving the care of patients with substance use problems:

*"Referral pathways, I think, function and operate at multiple layers and multiple levels from the very basic referral to the prescriber from the pharmacist themselves, leaving that up to the patient to take that up, to more formal forms of communication. The CPOP [Community Program for Opioid Pharmacotherapy] program is a terrific example of a collaboration between prescriber, dispenser, underpinned by a federal and state government funded and facilitated program . . . You've got a combination of a [PBS] Section 100 funded scheme administered by a state government body. . . and the prescribers and the dispensers are in contact on a daily, if not weekly, basis to manage their clients in an area of great social need. So whenever I'm trying to look for examples of where collaboration with multiple stakeholders—in particular, general practice and pharmacy—has been effective and where it can actually save lives, I think the CPOP program is a classic example."*

(P06 C)

The means of communication in the context of a referral also depended upon the nature and urgency of the referral, as clearly articulated by participant P07 C:

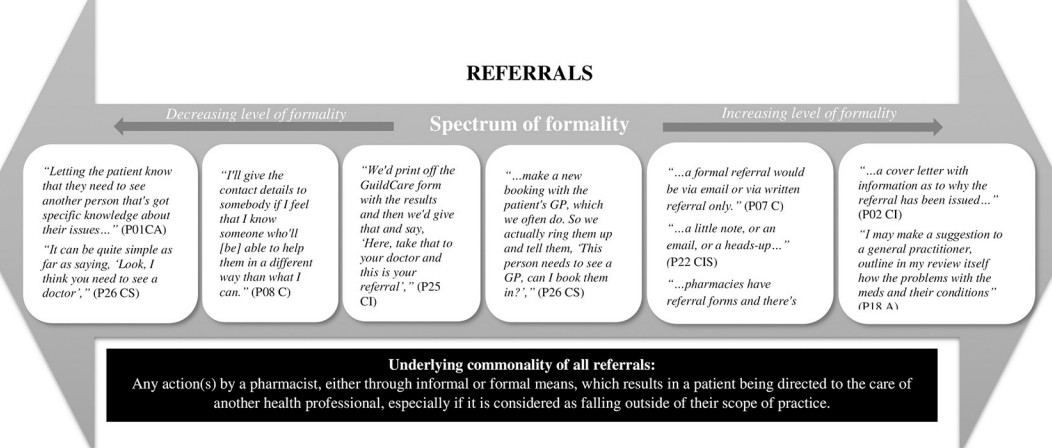

**Fig 1. Schematic representation of the nature of referrals by pharmacists on a spectrum of formality.**

"*So a referral is really when there's actually a clinical need, and then you identify the correct condition that needs to be referred to. It depends on the nature of it. If it's minor, it may be just the phone referral or a, 'Here's the address of the person. I suggest you get in contact with them.' But if there is actually a clinical need that's rather urgent, it's generally a formalised referral.*"

(P07 C)

For example, in instances where further clinical information had to be provided, referral with a written note was considered most appropriate:

"*If the blood pressure was consistently high upon many readings, the cover letter would be what I predict. . . based on their medications, and with the information of a graph or their readings attached to that as well.*"

(P02 CI)

In addition, the existing relationship with a practitioner was also seen as a factor influencing their choice of referrals:

"*It doesn't have to be written. It can be verbal. And when it's to someone who we have a very good relationship with, it's just verbal. We would make a note in our file.*"

(P04 C)

Written referrals were made using varying methods, including handwriting a referral note or creating an electronic referral letter following a specific template or using an existing software, such as the GuildCare [38] software, to assist with generation of the referral letters.

Although varying ways of referrals were described, there appeared to be an overarching commonality of all referrals mentioned by all participants. Overall, this could be summarised as any action(s) by a pharmacist, either through informal or formal means, which resulted in a patient being directed to the care of another health professional, especially if it was considered as falling outside of the scope of practice of the pharmacist.

### iii) Facilitators for effective interactions

Participants recognised the importance of pursuing and maintaining positive working relationships with other health professionals, including GPs and RACF staff. As was clearly articulated by one of the participants: "*. . .what did work, was building that really good relationship with our doctors. That worked for us in not only compounding, but in having that relationship with them in general.*" (P26 CS)

Four factors were identified to facilitate and encourage effective interactions: i) established rapport and relationship, ii) capitalising on existing workflow and resources, iii) location and proximity, and iv) experience and confidence (Table 2).

A key factor facilitating these interactions was identified as having an established rapport with the GPs and/or staff in the RACF:

"*A lot of the nursing homes that I work at have got a combined nurse practitioner/general practitioner collaborative model. And so, as part of that, I have very good relationships with the [nurses] because they're just so hungry for information, and they drive the process really well for us. They're a key enabler in getting medication reviews done in residential care.*"

(P10 A)

**Table 2. Facilitators and barriers to effective interactions with other health professionals.**

| Facilitators | Verbatim quotes |
|---|---|
| **Established rapport and relationship** | *"So you may get contacted by that other health professional to really have a chat. Say, for instance, diabetes from a diabetic educator, saying, 'Oh, look, what were the issues when you visited the patient?' So that's really again, it's a bit case-by-case sort of basis, but generally quite good and healthy relationships."* (P07 C)<br>*". . .because of the relationships, if there's any sort of issue or there's something that comes up, it's very easy just to address it. . ."* (P10 A)<br>*"The collaboration, it's some GPs [general practitioners]. . .who I know well now, they actually phone me about other things or email me about other things. So they sort of say, 'Look. Yes, I was worried about, yes, atrial fibrillation. . . and the new oral anticoagulants, the NOACs [novel oral anticoagulants],' and they might like some more information on that, which is totally unrelated to any review I've done for them, but they just want info. So it's created a dialogue between us, which is really good. And so it's a sign. This is a sign. Right? So we do collaborate. We do make time if they've got problems. But I talk to them if I have concerns, even while I'm in there. They'll bring things to me while I'm in the facility which are not related to my review at all, but you just do it because I want to be professional and they appreciate it. They want to consult."* (P18 A) |
| **Capitalising on existing workflow and resources** | *". . .I also go to the. . .they have another aged care independent people sort of facility and we meet up with them and talk about the wound care for their residents and clients."* (P14 CAS)<br>*"Our AHS [Aboriginal Health Service] used to host mornings for the health professionals to come over, so that was quite handy. So alternate Tuesdays, we'd have a physio would come over, and that would give me the opportunity to book a patient into the physio for that morning."* (P19 A)<br>*"I do have a lot of informal meetings. Sometimes I would, with individual GPs, just go out for like a lunch meeting or I'll just catch them between appointments."* (P20 C) |
| **Location and proximity** | *"The doctors next door we would meet with, I mean, we sort of see them quite regularly anyway. So if there's any need we'll have a chat about it then. But if there's anything coming up, for example, then one of us pharmacists will go next door and have a quick chat with the doctor . . ."* (P13 CAI)<br>*"I do a bit with exercise physiologists now. I've probably only started in the last six months or so. So one of the things we do is dealing with weight loss. If I feel as though they would benefit with knowing a little bit more of how to exercise, or they're also motivated to do stuff again. Yes. I've got a group of exercise physiologists probably 500 metres down the road. And another one probably a K [kilometre] and a half away or something. Between those two groups we send a bit there."* (P22 CIS)<br>*"So people see the value once you maybe talk to them or interact with them. . . I think it's more verbal with most of the other health professionals."* (P24 CAS) |
| **Experience and confidence** | *". . .credibility, because you've got someone backing you up who's got a very good reputation. . ."* (P15 CI)<br>*"They know that we're professional. They know they can ring us up. And we do get quite a lot . . .."* (P23 CAIS)<br>*"I think the collaborations have come because of the different components I work in, and I tend to have the collaborations as a result of the way I work. . . And because I've got that background where I do work in these practices, even if it's outside those hours, I'll just go and say to the staff, 'I'm going to go and see such and such. I'll just wait at their door.' So it's probably the confidence and also an acceptance that the practice is happy for me to do that because they know who I am."* (P24 CAS) |

| Barriers | Verbatim quotes |
|---|---|
| **Inadequate understanding and appreciation of roles** | *"I'm pretty disappointed in the lack of uptake by other health professionals; a lack of interest in medication reviews. I think it's very poorly understood, and unless there's an uptake driver for all participants, they won't participate."* (P06 C)<br>*". . . you can get GPs that are opposed to pharmacists becoming involved in their patient. . . they want full control of their patient rather having the patients see the pharmacists as well. Yes, that's a barrier if their attitude is not right . . ."* (P14 CAS)<br>*"We have made attempts to talk to doctors in the past, and it's not always—doctors seem to think that we're just trying to flog them something, and they don't really want to be talked to unless someone's buying them lunch."* (P15 CI)<br>*". . .you're still at the mercy of your prescriber. . . . . .so it's not only training of staff that you need but training of the prescriber."* (P22 CIS) |
| **Lack of a structured and consistent two-way referral pathway** | *"We have established referral pathways to psychologists and other user groups like WA Substance Users Group, Next Step. . . Are there really well-established referral pathways for our customer base? No, I don't believe there are. . . .in the vast majority of cases, we'd leave it up to the patient."* (P06 C)<br>*"It's mainly verbal. So just recommending them to see a doctor if, like you said, the treatment I've provided doesn't work or if I feel there's no adequate over-the-counter option. So, yes, mainly informal, verbal, but, for example, yesterday, actually, I typed up a letter because a patient was prescribed a medication that needed an authority to get it under PBS, or else it would have been quite expensive for them. So that one, I drafted up a letter to bring to the doctor at the next appointment."* (P12 CI)<br>*"Because it's all patient confidentiality. . . .. we don't have sort of a common integrated system. . ."* (P20 C) |

One participant further commented that it is important to *". . .show my face and keep that relationship. . ."* (P12 CI) to maintain the rapport with other health professionals.

It was also raised that understanding and capitalising on existing workflow which empowered operational efficiency in a practice setting enabled meaningful interactions with other health professionals, as described by one of the participants:

*". . .most of the [residential aged care] facilities, they'll have the GP visit on a certain date, or a certain day of the week, so if you want to work through the reports, then you just make sure that they have some time set aside to case conference those reports on the day [of] their visits."*

(P10 A)

Proximity between the pharmacy and other practices was also raised by some participants. Close proximity between the practices was identified as a facilitator to encourage interactions:

*"The one that's within walking distance from us, we see the doctors almost weekly just because they're coming in picking up scripts or picking up their own stuff."*

(P12 CI)

*"Having a psychologist next door really helped as well because I was able to bounce off ideas with him. And also with the exercise physiologist and the physio. . ."*

(P20 C)

One participant further commented that having the ability to meet with a GP face-to-face facilitated in-depth discussion and mutually beneficial interactions:

*"It gave us the opportunity to speak to those GPs that we would ordinarily speak to on the phone very briefly. It gave us an opportunity to speak a bit more in-depth with them. Which again, I think gained a bit of mutual respect, so we built that really good relationship with them."*

(P26 CS)

Besides system and external factors, the experience and confidence of the pharmacist was identified as having an impact on the establishment of effective interactions.

### iv) Barriers to effective interactions

Some participants reported mixed experiences and interactions with other health professionals. Whilst the importance of establishing effective interactions was acknowledged by all participants, two key barriers were identified: i) inadequate understanding and appreciation of roles and ii) lack of a structured and consistent two-way referral pathway (Table 2).

Many participants raised the issue regarding the lack of, or inadequate, understanding and appreciation of their professional roles by other health professionals, including GPs. In some instances, participants described the discord between their perception of their roles as pharmacists and that of a GP, which contributed to challenging interactions.

Another barrier was the lack of a structured and consistent two-way referral pathway. This was related to instances involving a pharmacist making a referral to, and receiving referrals or feedback from a prior referral, from another health professional, as articulated by P26 CS:

"*So the issue we have with the referrals, and I think we're still not anywhere close to getting a result from, it is getting the feedback from a referral. We occasionally see it when the patient brings back a script. But, something that we've always sort of said is that we would like to see the follow-up from it. So what was the result so that we're a bit more aware of that situation.*"

(P26 CS)

Specifically, it was raised that some GPs may not communicate or report back to the pharmacists when a pharmacist's recommendation as a result of a medication review had been taken up. The lack of a structured and consistent two-way referral pathway, to and from the GPs, was raised as a barrier to effective collaboration. One participant articulated the issue when conducting MedsChecks (in-pharmacy medication reviews):

"*Probably in about 30% of cases we would get an acknowledgement. Once again, if there's an incentive for them to do that, they tend to. If not, they don't.*"

(P06 C)

## v) Implementation of a national digital health record

It was evident from the interviews that participants recognised the importance of having a national, real-time, digital health record system to enable them to check a patient's health information when making a clinical recommendation. Specifically, participants commented on their views about the newly established national digital health record, My Health Record:

"*My perspective of the electronic health records is positive. I believe it's the way forward too because what it allows us to do is minimise medication-related incidences as well as to improve overall efficiency of the health-care system.*"

(P02 CI)

"*They make it easier for pharmacies or even other health professionals to actually know the whole record of the patients. The whole part of the history of everything or just make it easier for everyone to do the judgement.*"

(P03 C)

It was also considered by the participants that a national digital health record system was crucial in integrating with future infrastructure and processes:

"*I would consider it a key part of our infrastructure going forward, and so I think it's an absolute minimum requirement that we can access electronic healthcare records.*"

(P06 C)

"*I think it's absolutely crucial that electronic healthcare records become more widely implemented across the country and that pharmacists have the ability to interact with them.*"

(P10 A)

In support of the implementation of a national digital heath record system, one participant further emphasised the potential of such a system to assist with time management and improved efficiency for health professionals:

"*I think it's a great idea because it's a centralised database that, ideally, all healthcare practitioners can refer to and, I guess, add to. And I think it would free up the doctor's time instead of us bothering them while they're consulting. I think it'd be nice if you could just access it and answer some questions just by referring to this database.*"

(P12 CI)

Nevertheless, privacy was raised as a concern for some:

"*Personally, I'm concerned about it. I've tried going online myself to flag my record. . . to get an alert whenever anyone accesses My Health Record or when anyone cracks a pass but I haven't been able to do that personally. . . but I also have concerns that security will be breached and people's health records will become public.*"

(P13 CAI)

"*I'd imagine the privacy issue is going to be massive. . .*"

(P22 CIS)

Despite the issues raised, participants were confident that a national digital health record system would further facilitate inter-professional interactions and communication:

"*I think it should be mandatory because I think it would really help improve the healthcare system and reduce hospitalisation rates. . . and help health professionals to be better informed and can work together better because we get a lot of issues with doctors not talking to each other or specialists.*"

(P14 CAS)

"*. . .being able to get a comprehensive picture of what other health professionals are doing and how you're involved. It makes so much more sense to have that information available to everyone.*"

(P16 CI)

## Discussion

Pharmacists acknowledged the changing landscape of pharmacy practice, being driven by the needs and demands of both consumers and the Australian health system, as well as a perceived increased need for interactions with other health professionals mainly, but not limited to, GPs. All 26 pharmacists in this study had unique and varied qualifications and practice experiences including some in specialist services. This further highlighted changes in the scope of pharmacist practice towards specialisation to meet contemporary needs. This study has identified contemporary processes used to achieve effective communication and collaboration between pharmacists and other health professionals, including factors facilitating and inhibiting these interactions, and pharmacists' perspectives of the implementation of a national digital health record system.

To our knowledge, this is the first study that has identified pharmacists' varied interpretations of what constitutes a referral and found that referrals by pharmacists occurred on a spectrum of formality. This in itself is of interest as on one hand, it highlighted the lack of any consistent and structured referral pathway, or medium, but on the other, it divulged the

importance of a pharmacist's ability to employ varied strategies and judgements unique to a scenario. Regardless of the methods used to manage referrals, pharmacists shared a common underlying understanding of referrals being any action by a pharmacist, either through informal of formal means, which resulted in a patient being directed to the care of another health professional. This occurred in instances when a pharmacist determined that the request or scenario was beyond their scope of practice. This finding highlighted the significance of a pharmacist's understanding and an establishment of their own scope of practice. This is particularly timely considering the rapidly evolving practice landscape [1].

It was evident from this study there was a multidisciplinary approach to healthcare in the primary care setting, which can only be managed when effective communication, interaction and collaboration with other health professionals are established. This finding is consistent with previous national and international studies examining interactions between pharmacists and GPs and the impact of established rapport and positive working relationships [11, 39–43]. Although many forms of communication were identified, the results of the present study suggest that it was essential to have an established communication processes or systems, tailored to and considered on a case-by-case basis dependent on the pharmacist's existing rapport and location with the health professional and the nature, urgency and complexity of the scenario. Nevertheless, the fact that a variety of methods of referral was employed, which included those that were legally traceable versus verbal or informal notes made by pharmacists, further highlights the need for a structured referral process. The lack of a structured referral process may have resulted from the conventional role of pharmacists with dispensing being the major role and that when communication between a GP and a pharmacist occurred it was largely focused on prescription-related issues. This may no longer meet the contemporary partnership expected between these practitioners in the context of the current practice landscape when medication management is increasingly considered as a joint professional responsibility to ensure medicine safety and optimum patient care [14, 44].

The experience and confidence of the pharmacist emerged as one of the facilitators for effective interactions with other health professionals. The impact of pharmacist characteristics, including their determination, leadership skills, clinical expertise, rapport and having a proactive approach, have been previously demonstrated to exert a significant influence on the success of professional services delivery in the current practice landscape [6].

A study conducted in Spain by Rubio-Valera et al. [45], which evaluated factors influencing GPs and community pharmacists' collaboration, highlighted GPs' attitudes and perceptions of pharmacists to be fundamental to the establishment of collaboration. Therefore, the roles of pharmacists and their contribution to patient care should be clearly articulated and accepted to avoid potential miscommunication. A recent WA study also reported pharmacists' concerns about difficult working relationships with some GPs as one of the major barriers to allowing them to practise to their full scope [5]. These barriers further support the finding of this study that a frequent lack of a structured two-way referral pathway, which occurs for other health professionals, was detrimental to on-going collaboration. Considering an increased demand and need for primary care pharmacists, including community pharmacists, to provide triage services, the receipt of feedback and follow up from other health professionals throughout the transitions of care will become increasingly fundamental [14, 46–50]. It was identified in this study that scenarios involving the care of patients with substance use problems enrolled in opioid substitution programs often involved effective two-way interactions between the prescriber and the pharmacist. This active process may be linked to the fact that CPOP is a partially government-funded program and clear guidelines are available to facilitate collaboration [51].

This study identified that overall pharmacists were supportive of the establishment and implementation of a national digital health record system and considered the system would

enable further efficient direct and indirect collaboration with other health professionals through sharing of health-related information of a patient (with their consent). This topic has been extensively researched in other countries [52–56], but is a relatively recent initiative in Australia. The My Health Record was first launched in July 2012 (then referred to as the Personally Controlled Electronic Health Record), with government-coordinated participation trials conducted in 2016 [57], and the *My Health Record Guidelines for Pharmacists* published in June 2019 [27]. Although concerns about privacy were raised by some participants, it was not deemed to be a major barrier since processes are in place to maintain confidentiality, which is in line with findings from the trials [58].

Participants were purposively selected for this study, hence there may be bias in their opinion as they already had an interest in the topic. There may be bias from the researchers who were all pharmacists. However, strategies were in place to minimise any potential bias, including the inclusion of participants with diverse backgrounds, the employment of an independent person to contact potential participants and another independent person to conduct the interviews. Coding and analysis were also undertaken by two researchers to ensure reliability. The reported findings may not represent the views and perspectives of hospital pharmacists as they were not included. The present study focused on pharmacists practising in the primary care sector as it is acknowledged that pharmacists practising in the hospitals would likely have different exposures and experiences in relation to communicating and collaborating with other health professionals. All pharmacists interviewed were from WA and their views may not represent others who practise in countries with a different practice landscape.

## Conclusions

Pharmacists acknowledged the changing landscape of the Australian health system made communication and collaboration with other health professionals a more important element of current practice. A multidisciplinary approach to healthcare is contingent upon effective communication, interactions and relationships with other health professionals. Effort should be made to encourage the establishment of a consistent and structured two-way referral process between all health professionals across the continuum of care of a patient, possibly facilitated through the "My Health Record" Increased understanding and appreciation of the role and expertise of pharmacists at the primary care interface would further enhance inter-professional collaboration and improve quality use of medicines.

## Supporting information

**S1 File.**
(PDF)

## Acknowledgments

The authors would like to express their sincere gratitude to all the pharmacists for their participation in this study, as well as Mrs Bronwen Wright for administrative support in relation to communicating with participants and Mrs Patricia Filippin for conducting the interviews.

## Author Contributions

**Conceptualization:** Tin Fei Sim, H Laetitia Hattingh, Bruce Sunderland, Petra Czarniak.

**Data curation:** Tin Fei Sim, H Laetitia Hattingh, Bruce Sunderland, Petra Czarniak.

**Formal analysis:** Tin Fei Sim, H Laetitia Hattingh.

**Funding acquisition:** Tin Fei Sim, H Laetitia Hattingh, Bruce Sunderland, Petra Czarniak.

**Investigation:** Tin Fei Sim, H Laetitia Hattingh, Bruce Sunderland, Petra Czarniak.

**Methodology:** Tin Fei Sim, H Laetitia Hattingh, Bruce Sunderland, Petra Czarniak.

**Project administration:** Tin Fei Sim, Petra Czarniak.

**Resources:** Tin Fei Sim.

**Supervision:** Bruce Sunderland.

**Writing – original draft:** Tin Fei Sim.

**Writing – review & editing:** Tin Fei Sim, H Laetitia Hattingh, Bruce Sunderland, Petra Czarniak.

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
