## [Decision Letter · Decision Letter 0]

17 Apr 2020

PONE-D-20-04098

Effective communication and collaboration with health professionals: A qualitative study of pharmacists in the context of a changing practice landscape

PLOS ONE

Dear Dr. Sim,

Thank you for submitting your manuscript to PLOS ONE. After careful consideration, we feel that it has merit but does not fully meet PLOS ONE’s publication criteria as it currently stands. Therefore, we invite you to submit a revised version of the manuscript that addresses the points raised during the review process.

We would appreciate receiving your revised manuscript by Jun 01 2020 11:59PM. To enhance the reproducibility of your results, we recommend that if applicable you deposit your laboratory protocols in protocols.io, where a protocol can be assigned its own identifier (DOI) such that it can be cited independently in the future. For instructions see: http://journals.plos.org/plosone/s/submission-guidelines#loc-laboratory-protocols

We look forward to receiving your revised manuscript.

Kind regards,

Carl Richard Schneider, BN, BPharm (Hon), PhD

Academic Editor

PLOS ONE

2. Please include additional information regarding the interview guide used in the study and ensure that you have provided sufficient details that others could replicate the analyses. For instance, if you developed a guide as part of this study and it is not under a copyright more restrictive than CC-BY, please include a copy, in both the original language and English, as Supporting Information.

Reviewers' comments:

Reviewer's Responses to Questions

**Comments to the Author**

1. Is the manuscript technically sound, and do the data support the conclusions?

Reviewer #1: Yes

Reviewer #2: Yes

Reviewer #3: Yes

2. Has the statistical analysis been performed appropriately and rigorously? 

Reviewer #1: N/A

Reviewer #2: N/A

Reviewer #3: N/A

3. Have the authors made all data underlying the findings in their manuscript fully available?

Reviewer #1: Yes

Reviewer #2: Yes

Reviewer #3: Yes

4. Is the manuscript presented in an intelligible fashion and written in standard English?

Reviewer #1: Yes

Reviewer #2: Yes

Reviewer #3: Yes

5. Review Comments to the Author

Reviewer #1: The title should be more specific to help it be identified in data base searches. For example it should include that the study is in community pharmacists, in western Australia and that one of the main findings was that there is a need for a digital health record.

The language in the introduction could be improved. At the moment it is quite verbose and could be more direct. Additionally the grammar and syntax should be reviewed (see specific comments below). Try to avoid repetitive statements too e.g. Ling 299 ’It was notable’ is repeated above. It can be removed so that the sentence reads ‘Participants had….’

The conclusion in the abstract is not substantiated by the results presented. e.g. ‘across the continuum of patient care’ but the study was only done in primary care, so conclusions should not be drawn across the whole care spectrum.

The title and first paragraph refer to global contexts but the rest of the paper is quite specifically relation to Australian pharmacy practice. Consider making this explicit throughout and changing the title or update the content to include other policies, e.g. Summary Care Record as a digital health record in the UK.

The introduction could be improved by describing current knowledge of inter-professional collaboration and communication. There are some references to models currently available that should be introduced, e.g. https://bmjopen.bmj.com/content/bmjopen/6/3/e010488.full.pdf ; https://www.ncbi.nlm.nih.gov/pmc/articles/PMC4803863/ and https://link.springer.com/article/10.1007/s11096-017-0450-6 . The studies referred to in the discussion should be described in the introduciton. It could be stated that these models do not look at diverse enough groups of healthcare professionals (e.g. including nurses and psychologists) and so this would help the reader understand the aims of the study is too look at inter professional communication more broadly.

The methods are technically sounds and clearly reported.

The results are well documents. Table 2 may be unnecessary as data is also included in the pros.

Line 85: explain wha the pharmaceutical benefits scheme is. An international audience may be unfamiliar with this terminology.

Line 88 - 91: this is a very long sentence that is quite verbose. Try to split into two sentences or make it shorter and to the point.

Line 93: ‘has also been’ rather than ‘has been also’

Line 114: this paragraph could go a little further to provide some critical reflection on the introduction of My Health Record. What were the expectations? Preliminary evaluation? Lesson’s learned from other contexts?

Reviewer #2: Thank you for the opportunity to review this paper. I believe it touches an important topic. I have some suggestions that I believe will strengthen this paper:

Title

- As the focus of this manuscript is primary care pharmacists – I believe this should be stated in the title.

Introduction

- Due to the international audience, I think it would be helpful to expand what you mean by “Ongoing medicine price disclosures and reductions which commenced in 2007, have affected the Pharmaceutical Benefits Scheme [10, 11]. These changes, coupled with a positive trend in government funding for professional services through several Community Pharmacy Agreements [12]”. Define price disclosure and professional services that are funded. To allow for this extra space, I believe you can delete lines 74-81.

Method

- Why were only pharmacists in WA recruited?

- Can you expand on the purposive sampling. It is unclear how you prioritized some participants over others. I assume there would be hundreds of pharmacists meeting your inclusion criteria, it is not clear why some were chosen over others (e.g I am sure there are plenty of pharmacists that can administer immunization services, how did you choose which one to recruit?).

- Due to the international audience, you should define accredited pharmacist (perhaps in the intro is best for this).

- What is an independent interviewer? I am not sure what the relationship is, but if they were paid by the research team or a student, are they independent? You state you followed the COREQ item checklist, but I do not see you report on all 8 items for the Personal characteristics of the interviewer and researchers.

- Line 187 mentions COM-B model for the first time. This should be described for the reader. (I am also not clear how it was used in the results)

- Data saturation is not talked about in the method. Please add how this was defined.

- Was the data analysed independently by the 2 researchers?

- How was disagreement between the 2 researchers handled?

Results

- As this was purposive sampling, can you also add a table with the total participants demographics (not individual such as table 1). I cannot see if you have the spread that you were aiming for. E.g. if you were purposively aiming for equal gender, why is this not 50% for each gender. Same comment to see if there are equal distribution for roles, accreditation, specialized roles etc. It is hard to determine if saturation for each of these demographics was obtained if it is unclear how many pharmacists in each group participated.

- For ‘Means of communication and systems’ you discuss different practitioners that pharmacists communicate with. This does not seem to fit the theme ‘Means of communication’.

- The aim of your study is to ‘explore contemporary processes facilitating communication and collaboration between community pharmacists and other health professionals.’ A means of communication alone (e.g. having a telephone) does not seem to be a contemporary process to facilitate communication. I am not sure if your themes (i) means of communication, ii) referral processes, iv) barriers to effective interactions are relevant to your aim. Theme 3 and 5 might be the main answer to your aim and might benefit from being reanalysed with a clear answer to your aim.

- Line 251 states ‘Having an established communication means or system, which enables operational efficiency, was identified to have a major impact on the success of developing and maintaining an on going positive collaborative working relationships with other health professionals’. Why is this a separate theme to ‘Facilitators to effective interactions’. This comment might be linked to my comment on your aim.

- ‘Means of communication’ appears to just list different ways pharmacists communicate. Qualitative analysis should add some extra depth to this. Such as when one means of communication is preferred over another, to highlight the ‘why’ not just the ‘what’ (which could be answered in a survey). Further analysis of this theme may be required to identify the nuances of different modes of communication.

- The means of communication don’t seem particularly contemporary, as per your aim. Perhaps focus more on the contemporary processes instead of the traditional ones.

- I am not sure why ‘national digital health record’ is a separate Theme from ‘means of communication’. The national digital health record seems to be another way for practitioners to communicate.

Discussion

- There is a large focus on ‘referral’ in your discussion. However, this does not seem particular relevant to your aim of ‘contemporary processes facilitating communication and collaboration between community pharmacists and other health professionals.’ More emphasis should be placed on these contemporary processes (Maybe you need to define what these are e.g. would instant messaging apps like WhatsApp fit this?)

- Line 534 states “In this study, the lack of other health professionals’ understanding and/or appreciation and respect of pharmacists’ roles and expertise was identified as one of the major barriers to effective interactions.” As you did not interview other health professionals’ I do not believe you can make this claim.

Reviewer #3: Thank you for the opportunity to review this manuscript. At the time I was reading this, the COVID-19 pandemic had been in full swing for several weeks. The crisis and strain, particularly on primary care health professionals, highlight the importance of this study and supports the recommendations.

Please find below comments for your consideration.

As this is an Australian study it would be helpful to provide a paragraph about the Australian context for pharmacists – e.g. pharmacists don’t have prescribing rights, pharmacists are only just starting to work in non-prescribing roles in GP practices, there is no formal referral process in place such as is available to pharmacists in Scotland’s Minor Ailments Scheme, formal referral processes are in place for other health professionals.

Line 26

Suggest change

“the primary care setting requires an augmented need to communicate and collaborate with other…”

To

“the primary care setting augments the need to communicate and collaborate with other…”

Line 39 and 235 and 353

Suggest change “facilitators to effective interactions” to “facilitators for effective interactions” or “facilitators of effective interactions”

Line 40-42

Suggest change:

“Participants acknowledged the changing landscape of the Australian health system that affects communication and collaboration with other health professionals.”

to

“Participants acknowledged that the changing landscape of the Australian health system affects communication and collaboration with other health professionals.”

Line 87

Change has to have

Line 89

Change would to will (if you anticipate something it is in the future)

Suggest change to “.. by pharmacists will augment the need to interact and …”

Line 91

Add the word ‘to’

“healthcare professionals involved or referred to for the care of such patients”

Line 93

Change

“professional pharmacy services has been also identified in previous…”

To “professional pharmacy services has also been identified in previous…”

Line 120

Suggest change

“lead to systems that operate for improved interdisciplinary collaboration..”

To

“lead to systems that enhance interdisciplinary collaboration…”

Line 118

Change

“current processes leading to effective communication..”

To “current processes which lead to effective communication…”

Line 120

Suggest change to “lead to developing systems that enhance interdisciplinary collaboration and patient care.”

Line 134

I note that not all items of the COREQ checklist have been reported.

Line 147

Were community pharmacists, accredited pharmacists, and pharmacists with additional qualifications recruited? I associate accredited pharmacists as being accredited to provide HMRs and RMMRs through gaining additional qualifications – usually through AACP. Readers who are not Australian probably won’t make this connection. However I do not associate pharmacists who are accredited to immunise through additional training as accredited pharmacists.

In line 161 there appears to be a distinction between accredited pharmacists (Part D) and pharmacists with additional training (Part E).

Line 158

It would be useful to have access to the interview question guide.

Line 181

Was the analysis conducted by the two researchers independently or was it conducted collaboratively?

Line 184

Change “processes to effective “ to “processes for effective”

Line 187

What is the COM-B model? Is there a reference for this model?

Line 244

Change “that” to “who”

Line 253

Change “relationships” to “relationship”

Figure 1

This figure works well to illustrate the point.

Line 486-489

Suggest dividing into two sentences.

Line 499

Add “used to”

“methods they used to manage referrals”

Line 500

Change “actions” to “action”

Line 517

Change “referrals” to “referral”

Line 519-520

The grammar is not correct in this sentence.

I also do not see how the conclusion of GPs having power over pharmacists is derived.

Line 553

Change to

“This study identified that overall pharmacists were supportive of..”

Line 569

Change “personnel” to “person”

6. PLOS authors have the option to publish the peer review history of their article (what does this mean?). If published, this will include your full peer review and any attached files.

Reviewer #1: No

Reviewer #2: No

Reviewer #3: No

---

## [Author Response · Author response to Decision Letter 0]

28 May 2020

The authors would like to thank the editor and reviewers for reviewing the manuscript. We have considered all comments/suggestions and made necessary changes to improve the manuscript. As per the journal instructions, we have provided a detailed response to each point raised by the academic editor and reviewers, and attached the responses with this re-submission as a separate file. Please find attached separate file labelled "Response to Reviewers".

---

## [Editor Report · Decision Letter 1]

29 May 2020

Effective communication and collaboration with health professionals: A qualitative study of primary care pharmacists in Western Australia

PONE-D-20-04098R1

Dear Dr. Sim,

We are pleased to inform you that your manuscript has been judged scientifically suitable for publication and will be formally accepted for publication once it complies with all outstanding technical requirements.

With kind regards,

Carl Richard Schneider, BN, BPharm (Hon), PhD

Academic Editor

PLOS ONE

Additional Editor Comments (optional):

Thank you for your considered and comprehensive approach to revising the manuscript in response to reviewer comments. Well done!
---

## [Editor Report · Acceptance letter]

2 Jun 2020

PONE-D-20-04098R1 

Effective communication and collaboration with health professionals: A qualitative study of primary care pharmacists in Western Australia 

Dear Dr. Sim:

I'm pleased to inform you that your manuscript has been deemed suitable for publication in PLOS ONE. Congratulations! Your manuscript is now with our production department. 

Kind regards, 

on behalf of

Dr. Carl Richard Schneider 

Academic Editor

PLOS ONE